



# Mechanistic insights into chloroacetic acid production from atmospheric multiphase VOC-chlorine chemistry

Mingxue Li[1], Men Xia[2,3], Chunshui Lin[1], Yifan Jiang[1], Weihang Sun[1], Yurun Wang[1], Yingnan Zhang[1], Maoxia He[4], Tao Wang[1]

[1]Department of Civil and Environmental Engineering, The Hong Kong Polytechnic University, Hong Kong 999077, China
[2]Institute for Atmospheric and Earth System Research/Physics, Faculty of Science, University of Helsinki, Helsinki 00014, Finland
[3]Aerosol and Haze Laboratory, Beijing Advanced Innovation Center for Soft Matter Science and Engineering, Beijing University of Chemical Technology, Beijing 100029, China
[4]Environment Research Institute, Shandong University, Qingdao 266237, China

*Correspondence to*: Tao Wang (tao.wang@polyu.edu.hk)

**Abstract.** Chlorine-containing oxygenated volatile organic compounds (Cl-OVOCs) are indicators of atmospheric chlorine chemistry involving volatile organic compounds (VOCs). However, their formation mechanisms are insufficiently understood. Herein, a strong diel pattern of chloroacetic acid ($C_2H_3O_2Cl$) was observed with daytime peaks at 19 and 13 ppt (1-hour averages) in 2020 and 2021, respectively, at a coastal site in southern China. Ethene was previously proposed as the primary precursor responsible for daytime $C_2H_3O_2Cl$ levels, but a photochemical box model based on Master Chemical Mechanism (MCM) simulations indicates that ethene accounts for less than 1 %. Quantum chemical calculations suggest that other alkenes also can act as chloroacetic acid precursors. Using an updated gas-phase VOC-Cl chemistry model, we find that isoprene, the most abundant VOC at the sampling site, along with its oxidation products, accounts for 7 % of the observed $C_2H_3O_2Cl$ levels. Moreover, the simulation with the updated MCM produces appreciable levels of other Cl-OVOCs, especially chloro-acetaldehyde, a precursor of $C_2H_3O_2Cl$. We proposed the multiphase reaction of Cl-OVOCs to reconcile the overestimation of Cl-OVOCs and the underestimation of $C_2H_3O_2Cl$ in our gas-phase model. The estimated reactive uptake coefficients for various Cl-OVOCs range from $3.63 \times 10^{-5}$ to $2.34 \times 10^{-2}$, according to quantum chemical calculations and linear relationship modeling. Box model simulation with multiphase chemistry reveals that the heterogeneous conversion of chloro-acetaldehyde to $C_2H_3O_2Cl$ is a more important source of $C_2H_3O_2Cl$ than gas-phase reactions. Our study thus proposes a formation mechanism of gaseous $C_2H_3O_2Cl$ and highlights the potential importance of multiphase processes in VOC-Cl chemistry.

## 1 Introduction

Reactive halogen species play an important role in various atmospheric environmental processes, including the depletion of ozone ($O_3$) in the polar regions, the formation of secondary pollution in polluted areas, and global climate change (Saiz-Lopez et al., 2023; Simpson et al., 2015). Halogen radicals, such as Cl[•] and Br[•], primarily originate from the photolysis of photolabile halogen and deplete through the reactions with volatile organic compounds (VOCs) and $O_3$ in the earth's atmosphere (Lawler



et al., 2011; Osthoff et al., 2008; Spicer et al., 1998). The chemistry between atmospheric VOCs and reactive chlorine not only facilitates the formation of Cl˙-initiated secondary organic aerosols but also promotes the cycling of ˙OH-HO$_2$˙-RO$_2$˙, thereby enhancing the atmospheric oxidation capacity (Choi et al., 2020; Ma et al., 2023; Soni et al., 2023). A deeper understanding

of the chemistry between VOCs and Cl˙ is crucial for enhancing our knowledge of the atmospheric halogen cycle and its environmental impact.

Field measurements have detected several chlorine-containing aldehydes and ketones as indicators of atmospheric chlorine chemistry related to specific VOC species (Le Breton et al., 2018; Masoud et al., 2023). Moreover, concentrations of halogenated organic acids, including chloro-, bromo-, and iodo-acetic acids, have been measured in urban and coastal

atmospheres, with ethene suggested as a potential precursor for these compounds (Le Breton et al., 2018; Priestley et al., 2018; Xia et al., 2022; Yu et al., 2019). Chamber experiments examining gas-phase chlorine reactions with specific VOC precursors have identified these chlorine-containing aldehydes and ketones as products, as summarised in Tab. S1 (Blanco et al., 2010; Canosa-Mas et al., 2001; Kaiser et al., 2010; Orlando et al., 2003; Rodríguez et al., 2012; Wang et al., 2015; Wang and Finlayson-Pitts, 2001; Wennberg et al., 2018). These studies show that chloro-methylbutenone and 4-chloro-crotonaldehyde

are distinctive markers for the gas-phase chlorine chemistry of isoprene and butadiene, and inversely, formyl chloride, chloro-acetaldehyde, and chloroacetone can be derived from Cl˙ reactions with various VOC precursors, including isoprene, methyl vinyl ketone (MVK), and methacrolein (MACR). However, chloroacetic acid has not been reported in gas-phase chlorine reaction experiments involving alkenes; instead, organic acids have been proposed to originate from the ambient multiphase chemistry of aldehydes (Carlton et al., 2007; Franco et al., 2021). The precise generation pathways of these chlorine-containing

oxygenated VOCs (Cl-OVOCs), particularly chloroacetic acid, remain unclear.

In this study, field measurements of chloroacetic acid were conducted at a coastal site in Hong Kong during the autumn seasons of 2020 and 2021. The observed chloroacetic acid (and bromoacetic acid) was consistently concentrated around midday over the two years studied. We aimed to elucidate the potential chemical processes leading to chloroacetic acid production by performing quantum chemical (QC) calculations and updated chemical box model simulations, with important implications

for understanding the halogen chemistry in the atmosphere.

## 2 Methods

### 2.1 Field observations

Two field campaigns were performed from 6 October to 24 November 2020 and from 11 September to 1 November 2021 in a rural area (Cape D'Aguilar, 22.21° N, 114.25° E) at the southeast tip of Hong Kong Island, China. This site is affected by

anthropogenic activities in the nearby urban area and the Pearl River Delta, shipping activities in the nearby waters, biogenic emissions, and long-range transport from eastern China (Peng et al., 2022; Wang et al., 2019). Each field campaign simultaneously measured reactive halogens (C$_2$H$_3$O$_2$Cl, C$_2$H$_3$O$_2$Br, ClNO$_2$, Cl$_2$, HOCl, BrCl, and Br$_2$), trace gases (NO$_x$, N$_2$O$_5$, NH$_3$, CO, SO$_2$, and O$_3$), aerosol mass concentration (PM$_{2.5}$), aerosol surface area density ($S_a$), VOCs, NO$_2$ photolysis frequency





(jNO$_2$), and meteorological parameters (temperature (T) and relative humidity (RH)). Reactive halogens were measured using

an iodide-adduct time-of-flight chemical ionization mass spectrometer (I$^-$-Tof-CIMS, Aerodyne Research). The principles of

the I$^-$-Tof-CIMS have been described in detail (Lee et al., 2014). The reagent ions (I$^-$ and I(H$_2$O)$^-$) were produced by passing

1 Lpm of CH$_3$I-containing N$_2$ air through an inline ionizer ($^{210}$Po). The peaks of C$_2$H$_3$O$_2$Cl, C$_2$H$_3$O$_2$Br, ClNO$_2$, Cl$_2$, HOCl,

BrCl, Br$_2$, N$_2$O$_5$, and HONO were identified via high-resolution peak fitting and verification of the halogen isotopic ratios,

according to the natural isotopic abundances of Cl and Br. The mean mass resolution was ~4800. The detection limits were

70  1.71 ± 1.56 ppt for C$_2$H$_3$O$_2$Cl and 0.31 ± 0.26 ppt for C$_2$H$_3$O$_2$Br. The limits were defined as twice the standard deviation in

signals observed during background testing. In estimating the mixing ratio of C$_2$H$_3$O$_2$Br, the sensitivity ratio of C$_2$H$_3$O$_2$Br to

Br$_2$ was assumed to be the same as that of C$_2$H$_3$O$_2$Cl to Cl$_2$. Operational details of the CIMS and the measurement of other

parameters were given in our previous work (Xia et al., 2022), which reported appreciable levels of reactive bromines for our

2020 campaign.

**2.2 Quantum chemical calculations**

QC calculations are effective for investigating gas-phase and heterogeneous reaction mechanisms (Ma et al., 2018; Tentscher

et al., 2019; Xue et al., 2022) and have been adopted to complement field observations and model simulations (Yang, 2024).

In the present study, QC calculations were conducted to explore the Cl$^•$-initiated reaction mechanisms of alkenes, investigate

the multiphase process by which carbonyls convert into organic acids, and predict reactive uptake coefficients (γ) for Cl-

OVOCs. The Cl$^•$ addition of alkenes is a typical barrierless reaction without an available transition state. In determining the

reaction characteristics of Cl$^•$ + alkene reactions, relaxed potential energy surface scans along the dissociation paths of the Cl$^•$

addition intermediates were conducted to obtain the minimum energy paths; i.e., constrained optimization was performed at

each fixed C−Cl distance (Zhang et al., 2020). The structural optimization of Cl$^•$ addition intermediates and their relaxed scans

were performed at the M06-2X/aug-cc-pVTZ level of theory (Zhao and Truhlar, 2008a, b), which has been widely used to

study organic chlorine chemistry (Ma et al., 2021; Vijayakumar, 2021).

The key steps of gaseous carbonyl uptake in this work were the solvation of gaseous carbonyls, hydrolysis of aqueous

carbonyls, and evaporation of aqueous diols (an inverse reaction of gaseous diol solvation), which are referred to as diol

mechanisms (Franco et al., 2021). QC calculations provided the potential energy surface for the above processes. Geometry

optimizations and frequency calculations were conducted at the M06-2X/aug-cc-pVTZ level of theory. Gas-phase single-point

energies were calculated at the DLPNO-CCSD(T)/aug-cc-pVTZ level of theory with tightPNO and RIJK approximations.

Aqueous-phase Gibbs free energies were determined using the thermodynamic cycle method (Marenich et al., 2009; Tentscher

et al., 2019), along with solvation free energy at the M06-2X/6-31G(d)//SMD level of theory (Marenich et al., 2009).

Previous computational research has revealed that the solvation and hydrolysis kinetics of a trace gas in microdroplets are

related to their reactive uptake coefficients of the trace gas, as observed for N$_2$O$_5$ (Fang et al., 2024). We adopted the above

reaction energies for the solvation and hydrolysis of several OVOCs and their reported γ values to develop linear relationship



models, which can be used to predict the γ values of Cl-OVOCs. All QC calculations were performed using Gaussian 16 (Frisch et al., 2019) and ORCA (Neese, 2022) software.

### 2.3 Box model simulations

The Master Chemical Mechanism (MCM) describes detailed gas-phase degradation mechanisms of VOCs, including H-atom

abstraction reactions involving various alkanes and Cl$^\bullet$ (Jenkin et al., 1997, 2015; Saunders et al., 2003). In our previous studies, we enhanced the original model of gas-phase and heterogeneous chlorine and bromine chemistry by representing the addition reaction products of all alkenes with Cl$^\bullet$ as a dummy chlorine-containing RO$_2$$^\bullet$ radical, except for ethene and propene (Peng et al., 2021; Xia et al., 2022). To investigate the formation mechanism of chloroacetic acid, we refined the gas-phase chlorine chemistry of several key alkenes and incorporated the reactive uptake of Cl-OVOCs into the updated model. The

updates included (1) refining the rate constants and branching ratios for Cl$^\bullet$-initiated reactions involving typical alkenes; (2) adding the subsequent reaction mechanism of chlorine-containing RO$_2$$^\bullet$ radicals, assumed to be the same as the mechanism for the corresponding $^\bullet$OH reaction in the MCM; (3) adding the reactive uptake of Cl-OVOCs, whose coefficients for aerosols are predicted from QC calculations; and (4) adding the heterogeneous production and loss of chloroacetic acid.

All simulations were performed using Framework for 0-D Atmospheric Modeling (F0AM), version 4.3.0.1 (Wolfe et al.,

2016). The model was constrained every 5 minutes using the diurnal average concentrations of field-observed reactive halogens and relevant species in 2020 (Tab. S2). A 24-hour solar cycle for the campaign-averaged condition was simulated three times to stabilize intermediates, and the results of the last run were used for further analysis. We set up six scenarios (scenarios I–VI in Tab. 1) to explore the effects of each VOC precursor and the reactive uptake of carbonyls on chloroacetic acid generation.

### 3. Results and Discussion

#### 3.1 Field measurements of chloroacetic acid

The time series of the mixing ratios of C$_2$H$_3$O$_2$Cl and other meteorological parameters along with the correlation analysis of C$_2$H$_3$O$_2$Cl concentrations are depicted in Fig. 1. Daily averages of pollutant concentrations and meteorological parameters during the two observation periods are listed in Tab. 2. The mixing ratios of C$_2$H$_3$O$_2$Cl had a diurnal variation in both

observation periods. This variation was similar to the diurnal profiles for Cl$_2$, HOCl, BrCl, and Br$_2$. The average diurnal peaks for C$_2$H$_3$O$_2$Cl occurred at midday, with the maximum mixing ratios being 19 ppt in 2020 and 13 ppt in 2021. These values are approximately 5 and 3 times those observed in Manchester, United Kingdom during winter (Priestley et al., 2018). The levels of C$_2$H$_3$O$_2$Cl and inorganic reactive chlorine (denoted as Cl$_x$, where Cl$_x$ = 2 × Cl$_2$ + ClNO$_2$ + HOCl + BrCl) were higher in 2020 than in 2021.






The levels of $Cl_x$ had the strongest correlation with the $C_2H_3O_2Cl$ concentration, with the correlation coefficients being 0.74 and 0.84 in 2020 and 2021, respectively, indicating photochemical secondary formation of $C_2H_3O_2Cl$. $S_a$ and $jNO_2$ were positively correlated with the $C_2H_3O_2Cl$ level, suggesting that aerosols and solar radiation could be involved in $C_2H_3O_2Cl$ formation. Previous studies have revealed that nitrate photolysis on aerosols is an important source of atmospheric $Cl_2$ and $Br_2$

at the study site (Peng et al., 2022; Xia et al., 2022). We thus hypothesize that chloroacetic acid can be derived from photochemical processes relating to aerosols. We also found a stronger correlation between $S_a$ and $C_2H_3O_2Cl$ than between $S_a$ and $Cl_x$ (Fig. S1), suggesting that the $C_2H_3O_2Cl$ formation may directly involve heterogeneous reactions. The negative correlation between $C_2H_3O_2Cl$ and RH may be related to the removal of $C_2H_3O_2Cl$, such as reactive uptake onto clouds or aerosols.

**3.2 Gas-phase chlorine chemistry of alkenes**

The original MCM mechanism incorporates the formation mechanisms of chloroacetic acid with 1-chloroethane, 1,2-dichloroethane, and 1,2-dichloropropane as precursors. The specific mechanisms are depicted in Fig. S2. First, 1-chloroethane and 1,2-dichloroethane are oxidized by ˙OH to produce chloro-acetaldehyde, and 1,2-dichloropropane is oxidized to form chloroacetone. Subsequently, chloroacetic acid as a second-generation product is generated through the ˙OH oxidation of

chloro-acetaldehyde and photolysis of chloroacetone. However, to our knowledge, these three halocarbons have not been detected in field observations conducted in Hong Kong (Cao et al., 2023; Zeng et al., 2020).

Ethene has been proposed as a precursor of chloroacetic acid (Priestley et al., 2018) and was included in our previous work as part of the original model (Xue et al., 2015). However, simulation based on the original model shows that the ethene + Cl˙ reaction yields only 0.1 ppt of chloroacetic acid (<1 % of the observed value) while producing a higher concentration of Cl-

OVOCs, mainly containing chloro-acetaldehyde (at a maximum of 43 ppt), as plotted in Fig. S3.

To explain the observed elevated concentrations of chloroacetic acid, we propose gas-phase chlorine chemistry from alternative precursors to modify the original model, now referred to as the updated model. Due to differences in the reaction rate constants and branching ratios for each pathway involving alkenes and Cl˙ (NIST Chemical Kinetics Database, 2024), we performed QC calculations to reveal their reaction patterns through potential energy surface scans, as depicted in Figs. 2 and S4–S6. For

propene, Cl˙ preferentially adds to the terminal C (alpha-C) rather than middle C (beta-C). The alpha-C addition intermediate (IM1) of propene is lower in energy than the beta-C addition intermediate (IM2), and IM2 spontaneously converts to IM1 by overcoming a low energy barrier (~8 kcal mol$^{-1}$), as shown in Fig. 2. Isoprene, MVK, and MACR exhibit similar patterns (Figs. S3–S5). Our QC calculations indicate that alpha-C addition is the primary pathway for the alkene + Cl˙ reactions, with negligible beta-C addition. In contrast, previous chemical models only incorporate detailed reaction mechanisms for propene

and Cl˙ (including alpha-C and beta-C addition) and rough mechanisms for other alkenes (Xue et al., 2015).

Based on the above QC calculations, we refined the Cl˙-initiated reaction rate constants and branching ratios of alkenes (Tab. S3) and updated the gas-phase chlorine chemistry of several alkenes, as summarised in Fig. 3 (with the details for propene, isoprene, MVK, and MACR presented in Figs. S7–S10). The updated gas-phase mechanism reveals that chloro-acetaldehyde



and chloroacetone are first-generation products of other alkenes, excluding ethene, whereas chloroacetic acid serves as a second-generation product of these compounds. In particular, chloro-acetaldehyde and chloroacetone have been identified as ubiquitous products of gas-phase chlorine reactions of alkenes, such as isoprene, MVK, and MACR, in chamber experiments (Tab. S1). Additionally, propene is proposed as a precursor to chloroacetone (Wang et al., 2023).

We next investigated the contribution of several typical alkenes to chloroacetic acid using the updated gas-phase mechanism. The simulation is described as scenarios II–IV in the Methods section. The results show that the gas-phase chlorine chemistry of isoprene and its oxidation products accounts for 7 % of the observed mixing ratio of chloroacetic acid, and the role of propene is negligible (Fig. S3). The low yield of chloroacetic acid is explained as follows. The degradation of chloro-acetaldehyde is primarily driven by H-atom abstraction from the aldehyde group, leading to the formation of the acyl peroxy radical $CH_2ClCO_3^{\cdot}$. However, only a small fraction of $CH_2ClCO_3^{\cdot}$ is converted to chloroacetic acid through reactions with $HO_2^{\cdot}$ and $RO_2^{\cdot}$, whereas the majority of $CH_2ClCO_3^{\cdot}$ reacts with NO to form $CH_2ClO_2^{\cdot}$. The contribution of chloroacetone to $CH_2ClCO_3^{\cdot}$ is considered negligible because of its low photolysis rate constant in MCM.

Our gas-phase VOC-Cl model predicts the generation of up to 1 ppb of Cl-OVOCs (Fig. 4). Reactions of isoprene and its oxidation products (MVK and MACR) with $Cl^{\cdot}$ account for ~91 % of Cl-OVOCs, and ethene and propene play a minor role. Formyl chloride is the most abundant Cl-OVOC predicted with the model, followed by chloro-acetaldehyde and finally chloroacetone. All three Cl-OVOCs were identified in previous field observations of reactive chlorine species (Le Breton et al., 2018), but their concentrations were far lower than the level predicted using our model.

Our updated gas-phase alkene + $Cl^{\cdot}$ reactions support the production of various Cl-OVOCs but explain only 8 % of the observed chloroacetic acid concentration. Previous studies have reported that reactive uptake coefficients (γ) of several OVOCs (e.g., formaldehyde, glyoxal, methylglyoxal, acetone, 2-butanone, and 2,3-butanedione) on water or aerosol surfaces can reach levels of $10^{-5}$–$10^{-3}$ (De Haan et al., 2018; Iraci and Tolbert, 1997; Liggio et al., 2005; Schütze and Herrmann, 2004). Proposed multiphase mechanisms of the conversion of aldehydes to gaseous organic acids involve the multiphase equilibrium of aldehydes, diols, and acids, aldehyde hydrolysis, and gas- and aqueous-phase $^{\cdot}OH$ oxidations of diols, and they explain the production of formic acid from formaldehyde (Franco et al., 2021). Moreover, previous research has revealed heterogeneous reactions as an important source of formic acid (Jiang et al., 2023). We thus propose that our simulated Cl-OVOCs follow the same mechanism, and we estimate the contribution of multiphase processes to the formation of gaseous chloroacetic acid.

## 3.3 Multiphase production of chloroacetic acid

To address the overestimation of the chloro-acetaldehyde level and the underestimation of the chloroacetic acid level in our gas-phase chlorine model, we propose multiphase conversion mechanisms of chloro-acetaldehyde to gaseous chloroacetic acid and conducted QC calculations to determine the reaction potential energy surface (Fig. 5). Gaseous chloro-acetaldehyde could dissolve in the atmospheric condensed phase, and aqueous chloro-acetaldehyde then hydrates to form chloroethyl-diol. Finally, the diol could evaporate into the gas phase and react with $^{\cdot}OH$ to form chloroacetic acid. QC calculations indicate that chloroacetic acid readily forms from chloroethyl-diol undergoing $^{\cdot}OH$ oxidation and $O_2$ abstraction due to the low energy




barriers. However, the reaction energy barrier of chloro-acetaldehyde hydrolysis reaches 42.31 kcal mol$^{-1}$, indicating that the slow hydrolysis may be the rate-limiting step affecting the chloroacetic acid yield. Regardless, QC calculations support the possibility of the multiphase formation of chloroacetic acid.

To aid atmospheric models in simulating multiphase reactions of Cl-OVOCs, we conducted QC calculations to determine the γ values for chloro-acetaldehyde and other Cl-OVOCs, including formyl chloride and chloroacetone (see the Methods section). Briefly, we established linear relationships between the QC-calculated multiphase reaction energy and γ reported in previous experimental studies for several OVOCs, and we then used the derived relationships to predict the γ values of Cl-OVOCs. The model lgγ = –2.796 – 0.154$\Delta_r G_{hyd}$ has the largest correlation coefficient ($R^2$) of 0.727, where $\Delta_r G_{hyd}$ is the change in Gibbs free

energy for hydrolysis reactions (Fig. S11). The γ values predicted for formyl chloride, chloro-acetaldehyde, and chloroacetone are $2.34 \times 10^{-2}$, $8.23 \times 10^{-4}$, and $7.07 \times 10^{-5}$, respectively. The predictions for the other Cl-OVOCs are given in Tab. S4.

We then added the reactive uptake of Cl-OVOCs on ambient aerosol surfaces in our updated MCM model using λ determined in the above QC calculations. As shown in Fig. 6, the reactive uptake of Cl-OVOCs on aerosol surfaces reduces the maximum Cl-OVOC concentration from the ppb level to 49 ppt, contributing to a chloro-acetaldehyde loss of up to 95 % in our box

model simulation. The reactive uptake of Cl-OVOCs serves as an important sink for Cl-OVOCs and would resolve the overestimation of Cl-OVOC levels by our gas-phase model.

The proposed multiphase conversion mechanisms of chloro-acetaldehyde to chloroacetic acid are depicted in Fig. 7. They can be simplified as $CH_2ClCHO(g) + H_2O(l) \rightarrow CH_2ClCOOH(g)$ + other products. The first-order loss rate of chloro-acetaldehyde on aerosols is calculated using the γ value estimated above. To assess the contribution of heterogeneous processes to

chloroacetic acid formation, we estimated the yield (φ) of chloroacetic acid from chloro-acetaldehyde uptake as follows. A previous laboratory study shows a yield of 1 % – 2 % for oxalic acid from aqueous-phase photochemical reactions of glyoxal (Carlton et al., 2007). We assumed the yield (φ) of chloroacetic acid from chloro-acetaldehyde is twice the above process because the heterogenous production of chloroacetic acid involves one diol reaction whereas that of oxalic acid undergoes two diol reactions.

As discussed in Section 3.2, the gas-phase chlorine chemistry of alkenes accounts for only 8 % of the observed chloroacetate level. Including heterogeneous reactions increases the simulated level of chloroacetic acid to 24 %–48 % of the observed level. With these updates, the box model significantly improved its ability to simulate the chloroacetic acid level. However, there is still a discrepancy between the updated simulations and field measurements, which may result from uncertainty in the chloroacetic acid yield among other contribution factors. For example, in addition to alkenes, other VOCs of high molecular

weight, such as ethylbenzene, may serve as precursors of chloroacetic acid (Cui et al., 2021). Moreover, chloroacetic acid may be produced as a disinfection by-product from the chlorination of dissolved organic matter in the aqueous phase (Jahn et al., 2024).



## 4. Conclusion

A strong diel pattern of chloroacetic acid was observed at a coastal site in southern China during two photochemically active periods, indicating the intense Cl-initiated oxidation of VOCs. Chemical box model simulations based on the MCM indicated that ethene, previously proposed as a precursor of chloroacetic acid, contributed less than 1 % to the observed chloroacetic acid concentration. Even when considering other alkenes, their combined contribution would still only be 8 % of the observed value whereas the model predicts higher levels of Cl-OVOCs. We conclude that the underestimation of chloroacetic acid and overestimation of Cl-OVOCs can be attributed to the neglect of the heterogeneous processes involving Cl-OVOCs, as indicated by QC calculations. The QC calculations reveal the feasibility of the multiphase conversion of chloro-acetaldehyde to chloroacetic acid, and the reactive uptake coefficients of Cl-OVOCs including chloro-acetaldehyde are estimated to be $3.63 \times 10^{-5}$ to $2.34 \times 10^{-2}$. Adding the heterogeneous processes of these Cl-OVOCs to the MCM model can reduce the gaps between observed and simulated chloroacetic acid levels, implying potential multiphase formation mechanisms for chloroacetic acid and other secondary organics in the condensed phase. Future experimental studies are needed to validate the proposed multiphase chemical processes for Cl-OVOCs and establish a reliable yield of chloroacetic acid. In addition, it is important to investigate the implications of OVOC uptake for the fate of these compounds and the subsequent effects on secondary organic aerosols.

### Data Availability

All of the data used to produce this paper can be obtained by contacting Tao Wang (two.wang@polyu.edu.hk).

### Authors' Contributions

T. W. designed the field campaign and data analysis. M. L. conducted data analysis. M. X. and Y. J. conducted the field campaign. M. L. and T. W. prepared the manuscript with contributions from all co-authors.

### Competing Interests

At least one of the (co-)authors is a member of the editorial board of *Atmospheric Chemistry and Physics*.

### Acknowledgments

We are grateful to the Hong Kong Environmental Protection Department for providing the sampling locations and access to the VOC and trace gas data; to the Hong Kong Observatory for supplying the meteorological data; to the Hong Kong



Polytechnic University Research Facility in Chemical and Environmental Analysis for providing the ToF-CIMS. We are also very grateful to Qi Yuan for providing part of the VOC data and Fangfang Ma for revising our manuscript.

250 **Financial Support**

This research is supported by the Hong Kong Research Grants Council (Project No. T24-504/17-N and 15217922).

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





**Table 1. Model scenarios**

| Scenario | Precursor | | | Cl-OVOC uptake[a] | Heterogeneous formation of chloroacetic acid[b] |
|---|---|---|---|---|---|
| | $C_2H_4$ | $C_3H_6$ | $C_5H_8$ | | |
| I | √ | | | | |
| II | | √ | | | |
| III | | | √ | | |
| IV | √ | √ | √ | | |
| V | √ | √ | √ | √ | |
| VI | √ | √ | √ | √ | √ |

[a]The reactive uptake of Cl-OVOCs on the aerosols is based on estimated uptake coefficients derived from QC calculations, excluding the heterogeneous formation of chloroacetic acid. [b]The yield of chloroacetic acid from the reactive uptake of chloroacetaldehyde is estimated as twice that of oxalic acid from aqueous-phase photochemical reactions involving glyoxal (Carlton et al., 2007), and the γ value for chloroacetic acid is assumed to be the same as that for acetic acid (Wang et al., 2020).





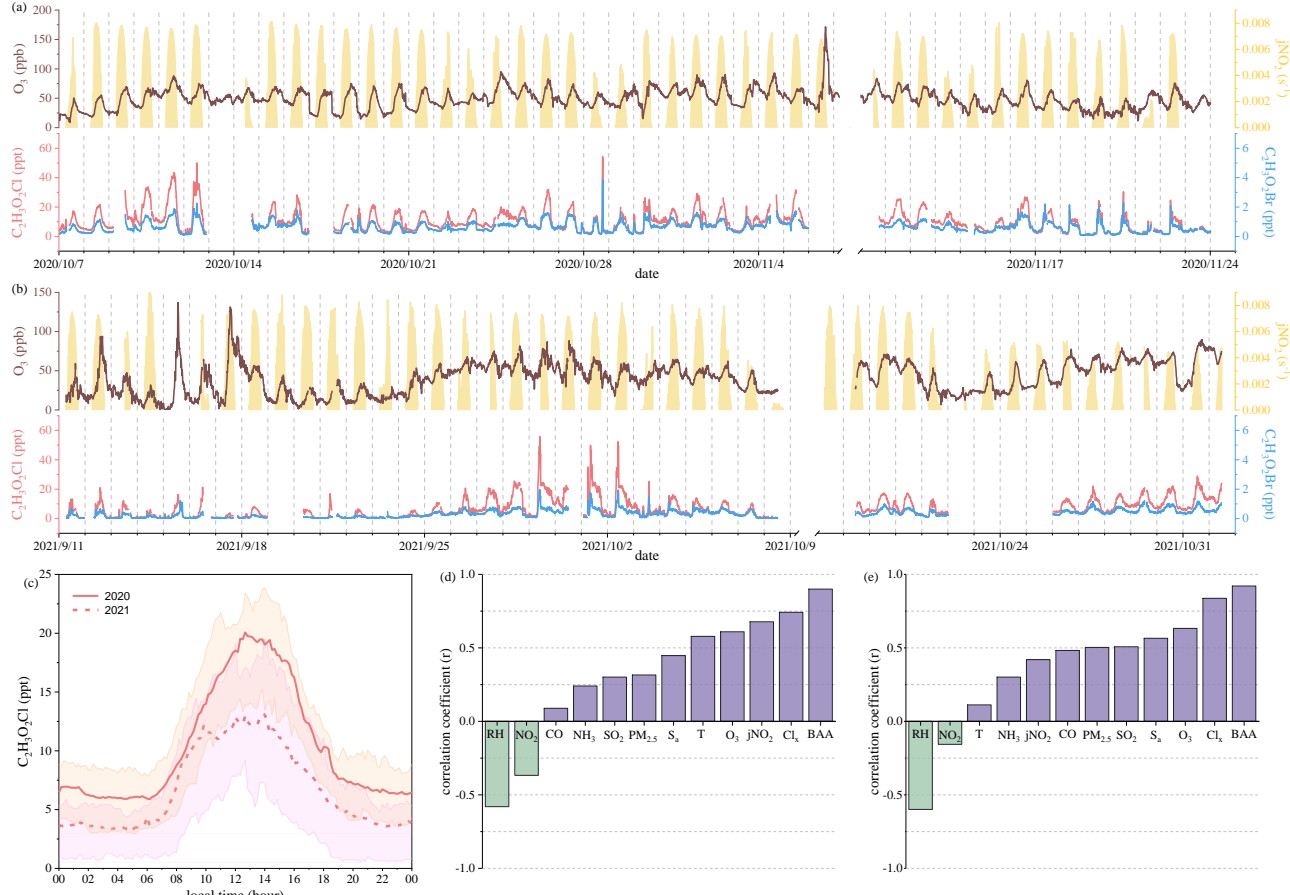


**Figure 1. Field observations of chloroacetic acid ($C_2H_3O_2Cl$) at a coastal site. Time series of the mixing ratios of $O_3$, $jNO_2$, $C_2H_3O_2Cl$, and $C_2H_3O_2Br$ (denoted as BAA) in (a) 2020 and (b) 2021. (c) Average diurnal variations of $C_2H_3O_2Cl$ in 2020 and 2021. The shaded areas represent 25 %–75 % of $C_2H_3O_2Cl$. Coefficients of correlation between $C_2H_3O_2Cl$, meteorological factors, and chemical constituents in (d) 2020 and (e) 2021. $Cl_x = 2 \times Cl_2 + ClNO_2 + HOCl + BrCl$. Data are 10-min averages in (a)–(c) and 1-h averages in**
**(d)–(e).**





**Table 2. Daily average values of pollutant concentrations and meteorological parameters**

| Parameter | 2020 | 2021 | Parameter | 2020 | 2021 |
|---|---|---|---|---|---|
| RH (%) | $75.0 \pm 11.0$ | $81.1 \pm 8.2$ | $ClNO_2$ (ppt)[b] | $491.2 \pm 388.1$ | $173.4 \pm 171.4$ |
| T (℃) | $24.0 \pm 1.8$ | $27.2 \pm 3.2$ | $Cl_2$ (ppt) | $15.4 \pm 16.4$ | $11.5 \pm 13.9$ |
| $jNO_2$ ($10^{-3}$ $s^{-1}$)[a] | $6.2 \pm 1.8$ | $5.3 \pm 2.4$ | HOCl (ppt) | $38.3 \pm 26.0$ | $31.5 \pm 35.9$ |
| $S_a$ ($\mu m^2$ $cm^{-3}$) | $178.5 \pm 81.9$ | $103.6 \pm 65.2$ | BrCl (ppt) | $0.62 \pm 0.56$ | $0.36 \pm 0.35$ |
| $PM_{2.5}$ ($\mu g$ $m^{-3}$) | $17.5 \pm 7.4$ | $8.8 \pm 6.3$ | $Br_2$ (ppt) | $3.0 \pm 1.9$ | $1.1 \pm 1.1$ |
| $SO_2$ (ppb) | $2.7 \pm 1.0$ | $5.1 \pm 0.5$ | $C_2H_3O_2Cl$ (ppt) | $10.1 \pm 6.9$ | $6.9 \pm 6.5$ |
| $O_3$ (ppb) | $49.6 \pm 16.5$ | $40.6 \pm 20.2$ | $C_2H_3O_2Br$ (ppt) | $0.65 \pm 0.36$ | $0.34 \pm 0.26$ |

[a]1-h average at midday; [b]average value at night (18:00–5:00 next day).

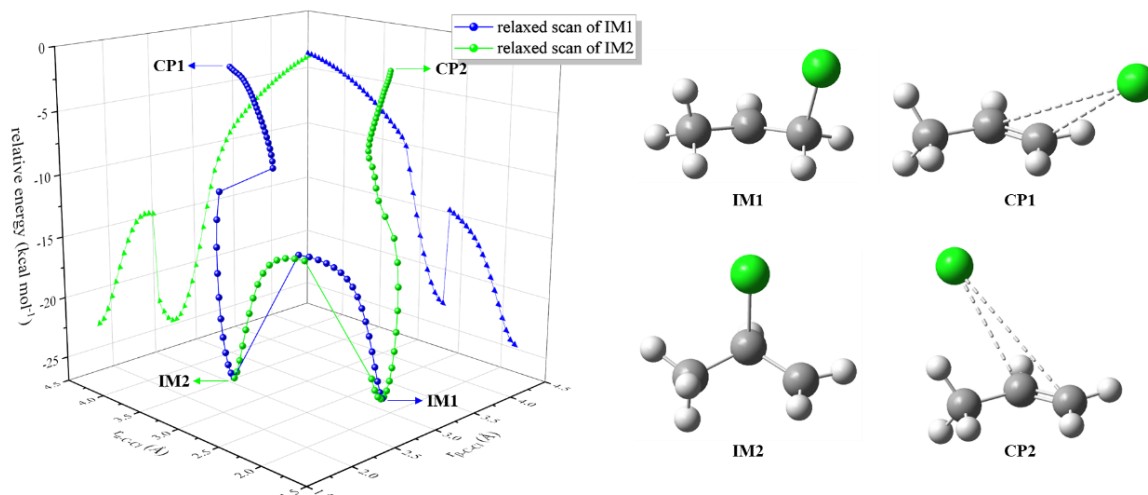


**Figure 2. Relaxed scan of Cl· addition to propene ($C_3H_6$). IM1 and IM2 are intermediates for Cl· addition to alpha-C and beta-C of propene, and CP1 and CP2 are the complexes derived from scans of IM1 and IM2 in terms of bond lengths ($r$) of alpha-C-Cl and beta-C-Cl as variables, respectively. Scanned potential energy surfaces of IM1 and IM2 take the total energy of the reactants Cl· + propene as zero for reference.**






**Figure 3. Gas-phase chlorine chemistry of typical alkenes. The MCM outlines the gas-phase generation of carbonyls, shown in black, and the Cl•-initiated reactions of alkenes are illustrated according to QC calculations in red. R and R' denote substituents. dec. = decomposition.**





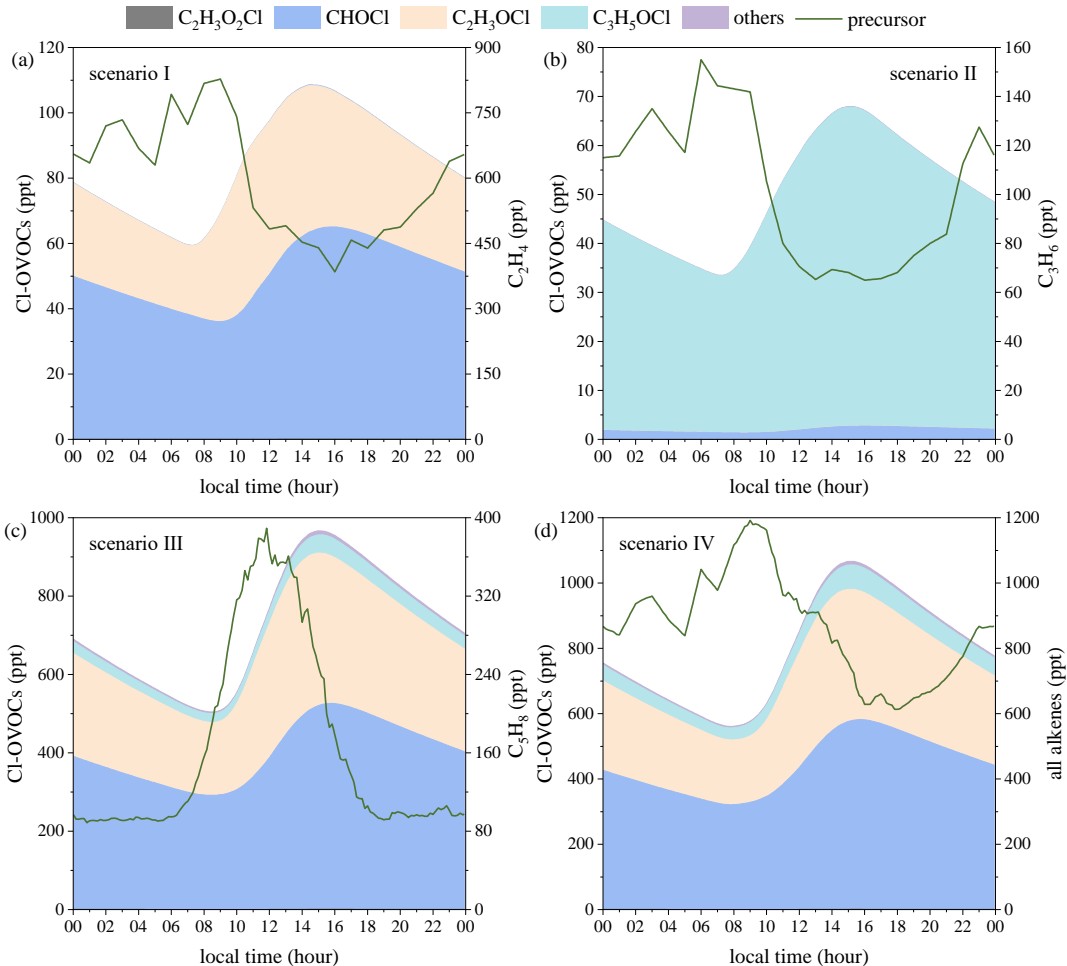


**Figure 4. Box model-simulated diurnal profiles of Cl-OVOCs generated from gas-phase VOC + Cl· reactions: (a)–(c) profiles for ethylene, propene, and isoprene, respectively, and (d) for all alkenes (the sum of the three alkenes). Scenarios I–IV are described in Tab. 1.**



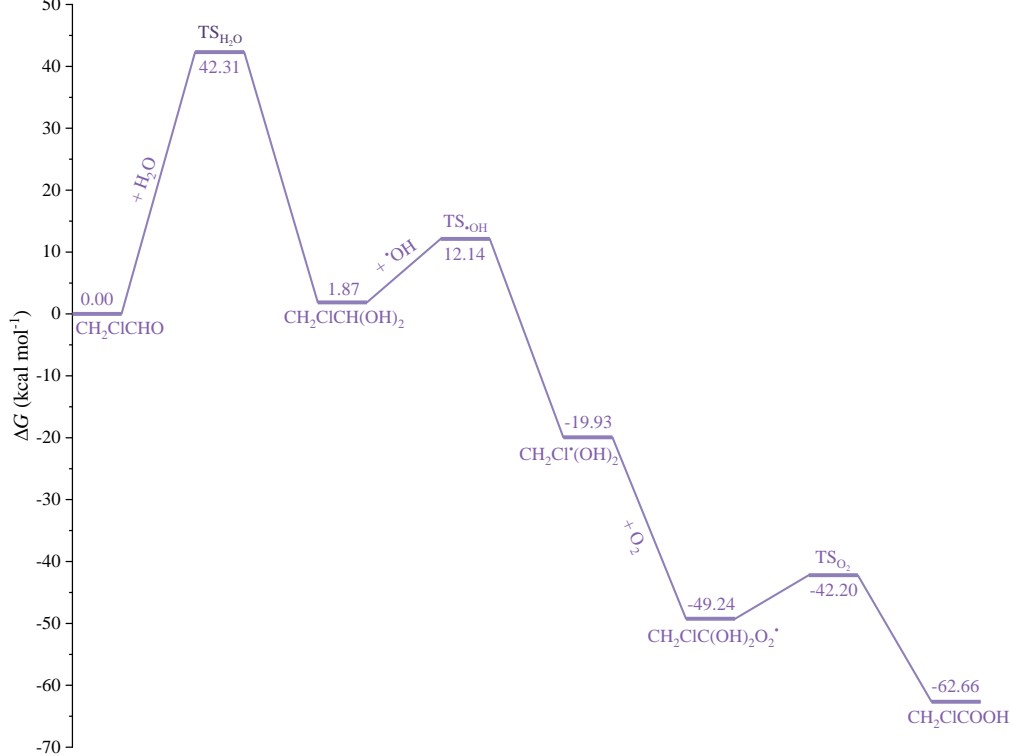


**Figure 5. QC-calculated potential energy surfaces of the aqueous-phase conversion of chloroacetic acid from chloro-acetaldehyde at 298 K. TS denotes the transition state connecting reactants and products.**

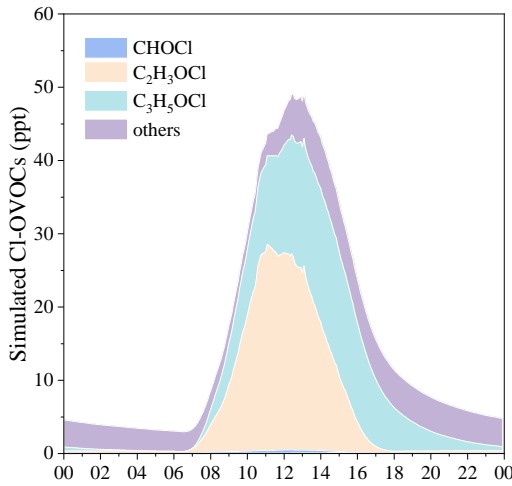

**Figure 6. Box model-simulated diurnal profiles of Cl-OVOCs generated from VOC + Cl· reactions with Cl-OVOC uptake on an aerosol surface (scenario V in Tab. 1).**





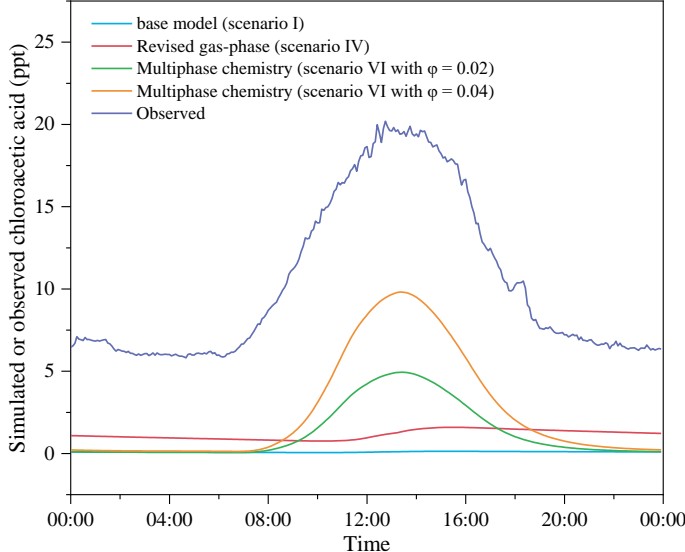

**Figure 7. Multiphase production of chloroacetic acid. The gas-phase conversion of chloro-acetaldehyde to chloroacetic acid, shown in black, is described in Fig. 3, whereas the multiphase reactions in red are presented in Fig. 5 according to QC calculations.**

**Figure 8. Comparison of measured and modelled diurnal profiles of chloroacetic acid.**