# Peer review of "Mechanistic insights into chloroacetic acid production from atmospheric multiphase VOC-chlorine chemistry"

_EGUsphere, 2024_

## Author Comment (AC1)

Dear Editor and reviewers,

Thank you very much for your comment on our manuscript "**Mechanistic insights into chloroacetic acid production from atmospheric multiphase VOC-chlorine chemistry (DOI: 10.5194/egusphere-2024-3137)**". We have cautiously considered all of the comments and responded as follows. Comments from the reviewers are shown in black *Italic* font. Response from the authors is shown in blue regular font. Revisions are shown in red regular font. The line numbers provided here refer to the ones in the revised manuscript.

*Referee #1:*

*Li et al. present observations of chloroacetic acid at a rural site in Hong Kong and showed that its formation cannot be fully explained by existing mechanisms. They propose two potential pathways for its production: (1) isoprene and its oxidation products, and (2) multiphase reactions. The draft is well-structured, and the results are clearly presented. However, I suggest the authors address the following comments before the manuscript is accepted for publication in ACP:*

Response: Thank you for your valuable suggestions. We have responded to your comments point by point below and added the discussion in the revised manuscript.

*1. The estimation of the reactive uptake coefficient relies on linear relationship models based on several OVOCs. I recommend evaluating the accuracy of this linear model and discussing the uncertainties associated with these uptake coefficients.*

Response: More regression evaluation indexes are provided to assess the accuracy of the linear relationship model. As shown in Fig. S11c, the linear relationship model between $\Delta_r G_{hyd}$ and lgλ exhibits the best correlation. The model's standard deviation (SD) is used for sensitivity testing of the predicted uptake coefficients (Fig. S12). We include a discussion of the effects of reactive uptake coefficients of Cl-OVOCs on simulated chloroacetic acid levels in the revised manuscript:

Lines 226 – 232: "A sensitivity analysis of the effects of reactive uptake coefficients of Cl-OVOCs on simulated chloroacetic acid levels using our updated model was also conducted (Fig. S12). Results show that a one standard deviation change (lgγ ± SD) in the predicted reactive uptake coefficients for Cl-OVOCs results in either a 5% increase or an 11% decrease in the simulated peak of chloroacetic acid. Our study underscores the significance of heterogeneous reactions in the production of chloroacetic acid and other organic acids, highlighting the necessity for more precise reaction parameters in future research."

[Figure]

Figure S11. Linear relationships between Gibbs free energy of diol reactions and reactive uptake coefficients of carbonyls. (a) $\Delta G_{sol}$ as the solvation energy of carbonyls; (b) $\Delta G^{\ddagger}_{hyd}$ and (c) $\Delta_r G_{hyd}$ as the Gibbs free energy barriers and changes in the hydration reactions of carbonyls; λ as the reactive uptake coefficients; $R^2$ as the coefficient of determination; RSS as the residual sum of squares; SD as the standard deviation.

[Figure]

Figure S12. Sensitivity testing of the effects of reactive uptake coefficients of Cl-OVOCs on simulated chloroacetic acid and Cl-OVOC levels using our updated model. $\lambda_{pred}$ are the predicted reactive uptake coefficients of Cl-OVOCs according to the linear relationship model between $\Delta_r G_{hyd}$ and $lg\lambda$, and SD is the standard deviation.

*2. The updated gas-phase chlorine chemistry and VOC-Cl model predict Cl-OVOCs concentrations up to 1 ppb, which significantly exceeds the observed values. The authors should provide an explanation for this large discrepancy.*

Response: In our simulation, the model considering only gas-phase VOC-Cl chemistry predicts high concentrations of Cl-OVOCs, mainly chloro-formaldehyde, chloro-acetaldehyde, and chloroacetone. The high yields of chloro-containing aldehydes and ketones predicted by our gas-phase model are consistent with the results of the gas-phase chamber experiment, but they do not match the field observations. To reconcile the overestimation of chloro-containing aldehydes and ketones and the underestimation of chloroacetic acid, we proposed multiphase chemistry including the reactive uptake of Cl-OVOCs and heterogeneous conversion of chloro-acetaldehyde to chloroacetic acid. A comparison of our gas-phase model simulation results with observations is presented in the last paragraph of section **3.2**:

Lines 175 – 178: "Our updated gas-phase alkene + Cl· reactions explain only 8% of the observed chloroacetic acid concentration and significantly overestimate Cl-OVOCs. The photochemical formation of chloroacetic acid and Cl-OVOCs results in their simulated diurnal cycles of daytime increase and nighttime decrease, failing to replicate the observed patterns. The inconsistency between simulated and observed results of chloroacetic acid and Cl-OVOCs implies possible missing sinks or sources."

*3. The inclusion of both updated gas-phase chemistry and heterogeneous reactions increases the simulated levels of chloroacetic acid by 32–56%. I suggest adding a discussion on the potential role of other missing mechanisms to account for the remaining gap.*

Response: Other missing sources have been discussed in the last paragraph of section **3.3**.

Lines 232 – 242: "With the aforementioned updates, the box model significantly improved its ability to simulate chloroacetic acid. However, there is still a discrepancy between the updated simulations and field measurements, which may result from the uncertainty in the parameters we use and other factors affecting chloroacetic acid. For example, previous studies have reported the RH dependence of the reactive uptake coefficients of aldehydes and

organic acids (Chen et al., 2021; De Haan et al., 2018; Gen et al., 2018; Tong et al., 2010; Zeineddine et al., 2023). Other reactions of aldehydes in aerosols such as ·OH oxidation, sulfite addition and Maillard-like reactions with reduced nitrogen species could compete with hydrolysis (Shen et al., 2024; Tang et al., 2022), potentially suppressing the yield of organic acids from the multiphase conversion of aldehydes. α-Chloro-β-ketones such as chloroacetone may also contribute to chloroacetic acid formation through atmospheric heterogeneous chemistry, supported by our QC calculations. In addition to alkenes, other VOCs of high molecular weight, such as ethylbenzene, may serve as precursors of chloroacetic acid (Cui et al., 2021). Moreover, chloroacetic acid may be produced as a disinfection by-product from the chlorination of dissolved organic matter in the aqueous phase (Jahn et al., 2024)."

*4. Line 193: Considering the slow rate of hydrolysis reactions, how do QC calculations support the plausibility of chloroacetic acid formation via multiphase processes? This result appears contradictory.*

Response: Our original manuscript investigates only the hydrolysis of OVOCs with water monomer to compare the relative energies of their hydrolysis thereby estimating the reactive uptake coefficients of Cl-OVOCs through QC calculations and linear relationship models. Considering the aqueous and acidic nature of aerosols, we investigated the hydrolysis of OVOCs with water dimer ($(H_2O)_2$), water trimer ($(H_2O)_3$), and sulfuric acid ($H_2SO_4$) in our revised version (Fig. 5). Results showed that multiple water molecules or acid catalysts involved in the hydrolysis reaction significantly reduce the energy barrier. Our QC calculations in the revised manuscript support the plausibility of chloroacetic acid formation via multiphase processes. The revised manuscript is as follows:

Lines 193 – 200: "The calculated energy barrier of chloro-acetaldehyde hydrolysis with a water molecule reaches 37.5 kcal mol$^{-1}$, which is reduced by the water dimer and trimer to 25.2 and 21.3 kcal mol$^{-1}$, respectively. Considering the acidic nature of aerosol, a molecule of sulfuric acid catalyzes chloro-acetaldehyde hydrolysis with an energy barrier of 6.1 kcal mol$^{-1}$. Multiple water molecules or acid catalysts involved in the hydrolysis reaction significantly lower the energy barrier, indicating the rapid hydrolysis feasibility of chloro-acetaldehyde in atmospheric aerosols. QC calculations also indicate that chloroacetic acid readily forms from chloroethyl-diol undergoing ·OH oxidation and $O_2$ abstraction due to the low energy barriers in both gas and aqueous phases, as depicted in Fig. 5b. The QC calculations support the plausibility of the multiphase formation of chloroacetic acid."

[Figure]

Figure 5. QC-calculated potential energy surfaces of the multiphase conversion of chloroacetic acid from chloro-acetaldehyde at 298 K. (a) The hydrolysis potential energy surfaces of chloro-acetaldehyde with water monomer ($H_2O$), water dimer (($H_2O)_2$), water trimer (($H_2O)_3$), and sulfuric acid ($H_2SO_4$) in aqueous phase; (b) potential energy surfaces of the conversion of chloroethyl-diol to chloroacetic acid in gas and aqueous phases. TS denotes the transition state connecting reactants and products, RC and PC denote reactant complexes and product complexes.

*5. Lines 131–134 and Figure S1: The correlation coefficient between $S_a$ and $C_2H_3O_2Cl$ is not strong enough to indicate a robust correlation. Please address this limitation.*

Response: The correlation coefficients for $S_a$ and $C_2H_3O_2Cl$ levels were 0.44 and 0.54 in 2020 and 2021, respectively, which are "moderately" correlated. The inappropriate statement below has been corrected or removed from the revised manuscript.

Lines 128 – 130: "Daytime $S_a$ and $jNO_2$ were moderately positively correlated with the $C_2H_3O_2Cl$ level (Fig. S1), suggesting that aerosols and solar radiation could be involved in $C_2H_3O_2Cl$ formation."

""

*6. Figure 1: Why are the observed $C_2H_3O_2Cl$ levels higher in 2020 compared to 2021? Can this discrepancy be explained by the proposed mechanisms?*

Response: Our proposed mechanism for chloroacetic acid formation indicates that reactive chlorine concentrations, precursor levels (primarily isoprene), and heterogeneous reaction processes impact the simulated levels of chloroacetic acid. The higher concentrations of observed chloroacetic acid in 2020 compared to 2021 can be partially attributed to elevated reactive chlorine species despite lower average isoprene concentrations in 2020.

Complex components of the aerosol would also influence the formation of chloroacetic acid. For example, higher $SO_2$ concentrations in 2021 compared to 2020 (Tab. 2) may suppress the hydrolysis reaction of aldehydes in aerosols, thereby reducing chloroacetic acid production. The limitations of our updated model are discussed:

Lines 236 – 238: "Other reactions of aldehydes in aerosols such as ˙OH oxidation, sulfite addition and Maillard-like reactions with reduced nitrogen species could compete with hydrolysis (Shen et al., 2024; Tang et al., 2022), potentially suppressing the yield of organic acids from the multiphase conversion of aldehydes."

*7. Figure 4: The model-simulated diurnal cycle of Cl-OVOCs does not match the observed diurnal pattern of $C_2H_3O_2Cl$ shown in Figure 1. Please explain this inconsistency.*

Response: The simulated diurnal cycle of chloroacetic acid using our gas-phase model does not match the observed diurnal pattern, which is modified by adding heterogeneous sources and sinks. A comparison of our gas-phase model simulation results with observations is presented in the last paragraph of section **3.2**:

Lines 175 – 178: "Our updated gas-phase alkene + Cl˙ reactions explain only 8% of the observed chloroacetic acid concentration and significantly overestimate Cl-OVOCs. The photochemical formation of chloroacetic acid and Cl-OVOCs results in their simulated diurnal cycles of daytime increase and nighttime decrease, failing to replicate the observed patterns. The inconsistency between simulated and observed results of chloroacetic acid and Cl-OVOCs implies possible missing sinks or sources."

*8. Figure 4: The gray shading representing $C_2H_3O_2Cl$ is not visible in the plot. Please clarify or correct this issue.*

Response: Compared to other Cl-OVOCs, very low concentrations of $C_2H_3O_2Cl$ could not be visualized in Fig. 4. $C_2H_3O_2Cl$ was removed from Fig. 4 and the caption added "The diurnal profiles of $C_2H_3O_2Cl$ (maximum 1.6 ppt) are not visible".

**Reference**

Chen, X., Zhang, Y., Zhao, J., Liu, Y., Shen, C., Wu, L., Wang, X., Fan, Q., Zhou, S., and Hang, J.: Regional modeling of secondary organic aerosol formation over eastern China: The impact of uptake coefficients of dicarbonyls and semivolatile process of primary organic aerosol, Science of The Total Environment, 793, 148176, https://doi.org/10.1016/j.scitotenv.2021.148176, 2021.

Cui, H., Chen, B., Jiang, Y., Tao, Y., Zhu, X., and Cai, Z.: Toxicity of 17 Disinfection By-products to Different Trophic Levels of Aquatic Organisms: Ecological Risks and Mechanisms, Environ. Sci. Technol., 55, 10534–10541, https://doi.org/10.1021/acs.est.0c08796, 2021.

De Haan, D. O., Jimenez, N. G., De Loera, A., Cazaunau, M., Gratien, A., Pangui, E., and Doussin, J.-F.: Methylglyoxal uptake coefficients on aqueous aerosol surfaces, J. Phys. Chem. A, 122, 4854–4860, https://doi.org/10.1021/acs.jpca.8b00533, 2018.

Gen, M., Huang, D. D., and Chan, C. K.: Reactive Uptake of Glyoxal by Ammonium-Containing Salt Particles as a Function of Relative Humidity, Environ. Sci. Technol., 52, 6903–6911, https://doi.org/10.1021/acs.est.8b00606, 2018.

Jahn, L. G., McPherson, K. N., and Hildebrandt Ruiz, L.: Effects of relative humidity and photoaging on the formation, composition, and aging of ethylbenzene SOA: Insights from chamber experiments on chlorine radical-initiated oxidation of ethylbenzene, ACS Earth Space Chem., 8, 675–688, https://doi.org/10.1021/acsearthspacechem.3c00279, 2024.

Shen, H., Huang, L., Qian, X., Qin, X., and Chen, Z.: Positive Feedback between Partitioning of Carbonyl Compounds and Particulate Sulfur Formation during Haze Episodes, Environ. Sci. Technol., 58, 21286–21294, https://doi.org/10.1021/acs.est.4c07278, 2024.

Tang, S., Li, F., Lv, J., Liu, L., Wu, G., Wang, Y., Yu, W., Wang, Y., and Jiang, G.: Unexpected molecular diversity of brown carbon formed by Maillard-like reactions in aqueous aerosols, Chem. Sci., 13, 8401–8411, https://doi.org/10.1039/D2SC02857C, 2022.

Tong, S. R., Wu, L. Y., Ge, M. F., Wang, W. G., and Pu, Z. F.: Heterogeneous chemistry of monocarboxylic acids on $\alpha$-$Al_2O_3$ at different relative humidities, Atmos. Chem. Phys., 10, 7561–7574, https://doi.org/10.5194/acp-10-7561-2010, 2010.

Zeineddine, M. N., Urupina, D., Romanias, M. N., Riffault, V., and Thevenet, F.: Uptake and reactivity of acetic acid on Gobi dust and mineral surrogates: A source of oxygenated volatile organic compounds in the atmosphere?, Atmos. Environ., 294, 119509, https://doi.org/10.1016/j.atmosenv.2022.119509, 2023.

---

## Author Comment (AC2)

Dear Editor and reviewers,

Thank you very much for your comment on our manuscript "**Mechanistic insights into chloroacetic acid production from atmospheric multiphase VOC-chlorine chemistry (DOI: 10.5194/egusphere-2024-3137)**". We have cautiously considered all of the comments and responded as follows. Comments from the reviewers are shown in black *Italic* font. Response from the authors is shown in blue regular font. Revisions are shown in red regular font. The line numbers provided here refer to the ones in the revised manuscript.

*Referee #2:*

*The manuscript "Mechanistic insights into chloroacetic acid production from atmospheric multiphase VOC-chlorine chemistry" by Li et al. describes the potential heterogeneous processes of chloroacetic acid production in observations at Hong Kong by performing quantum chemical calculations and chemical box model simulations. The formation mechanism of chloroacetic acid in the atmosphere is poorly understood, as such, I feel this study, which provides a comprehensive assessment of this chemistry, is of interest to the community and falls within the scope of ACP. The manuscript is also well written and organized. I recommend publication after addressing the following points.*

Response: Thank you for your encouraging comments and valuable suggestions. Please find our point-by-point response below.

*1. Page 3, line 93-: I am not very sure whether a linear relationship analysis between hydrolysis of OVOCs and their reported γ values could be used to predict γ of Cl-VOCs. For $N_2O_5$ in liquid water maybe it is fine as its bulk hydrolysis rate is quite fast, and there are no other significant processes in pure liquid water. But here, the hydrolysis rates of Cl-VOCs are slow, oxidation in aqueous phase also plays a role, and the composition in aerosol water is really complicated. Please discuss whether these factors would have an impact.*

Response: Considering the aqueous and acidic nature of aerosols., we reinvestigated the hydrolysis of chloro-acetaldehyde by water dimers, water trimers and sulfuric acid. As shown in revised Fig. 5, the energy barrier for the hydrolysis of chloro-acetaldehyde to form diol by water trimer is reduced to 21.3 kcal mol$^{-1}$, which means that this reaction rate reaches $2.8 \times 10^{-3}$ s$^{-1}$ based on the TST theory. Even one molecule of sulfuric acid catalyzes the hydrolysis of chloro-acetaldehyde with an energy barrier of 6.1 kcal mol$^{-1}$ and a reaction rate constant of $2.8 \times 10^8$ s$^{-1}$. The reaction rate for the $^{\bullet}$OH oxidation of chloro-acetaldehyde is $10^{-6} – 10^{-3}$ s$^{-1}$, which was estimated based on the $^{\bullet}$OH concentrations in the aerosol ($[^{\bullet}OH]_{aq}$) as $10^{-15} – 10^{-12}$ M in the tropospheric aqueous phase (Herrmann et al., 2010) and the reaction rate constant as $3.82 \times 10^9$ M$^{-1}$ s$^{-1}$ (Huang et al., 2021). The $^{\bullet}$OH oxidation of chloro-acetaldehyde may compete with hydrolysis in marine areas with high $[^{\bullet}OH]_{aq}$. The fate of chloro-acetaldehyde in aerosols depends on the aerosol components, such as concentrations of $^{\bullet}$OH and sulfuric acid. In our revised manuscript, QC calculations support the possibility of multiphase formation of chloroacetic acid, and the competing reactions of aldehyde hydrolysis in aerosols are mentioned:

Lines 236 – 238: "Other reactions of aldehydes in aerosols such as $^{\bullet}$OH oxidation, sulfite addition and Maillard-like reactions with reduced nitrogen species could compete with hydrolysis (Shen et al., 2024; Tang et al., 2022), potentially suppressing the yield of organic acids from the multiphase conversion of aldehydes. "

[Figure]

Figure 5. QC-calculated potential energy surfaces of the multiphase conversion of chloroacetic acid from chloro-acetaldehyde at 298 K. (a) The hydrolysis potential energy surfaces of chloro-acetaldehyde with water monomer ($H_2O$), water dimer (($H_2O)_2$), water trimer (($H_2O)_3$), and sulfuric acid ($H_2SO_4$) in aqueous phase; (b) potential energy surfaces of the conversion of chloroethyl-diol to chloroacetic acid in gas and aqueous phases. TS denotes the transition state connecting reactants and products, RC and PC denote reactant complexes and product complexes.

*2. It seems that the method for deriving γ also assumes that gas-phase diffusion limitations were negligible, is this true? Please clarify.*

Response: The resistor model provided a concise formulation for estimating the reactive uptake coefficient, γ, of the trace gas (Davidovits et al., 2011). In the model, the elemental steps of heterogeneous uptake kinetics include gas-phase diffusion, mass accommodation, and solubility/reactivity in liquids, as expressed:

$$\frac{1}{\gamma} = \frac{1}{\Gamma_{dif}} + \frac{1}{\alpha} + \frac{1}{\Gamma_{rxn} + \Gamma_{sat}}$$

where $\Gamma_{dif}$ is the gas-phase diffusion limitation, α is the mass accommodation coefficient, $\Gamma_{rxn}$ and $\Gamma_{sat}$ describe the liquid-phase reaction and solubility processes.

For typical submicrometer-sized aerosol particles in the atmosphere, gas-phase diffusion does not usually limit uptake coefficients unless the uptake coefficient is large. Previous studies have also reported that gas-phase diffusion was not the primary limitation on observed uptake coefficients at RH ⩽ 98% (De Haan et al., 2018). So, we assume that the gas-phase diffusion limitations of OVOCs were negligible.

By the way, mass accumulation coefficients for formaldehyde, acetaldehyde and glyoxal were reported as 0.02, < 0.03 and 0.023, respectively (Jayne et al., 1992; Schweitzer et al., 1998). Considering their uptake coefficients at

the $10^{-3}$ level, we infer that interfacial mass accumulation is also not a limiting factor for the reactive uptake of aldehydes.

Solution process and chemical reactions in the liquid phase may dominate the uptake of aldehydes. We therefore used the free energy of dissolution and the hydrolysis reaction energy difference of OVOCs to build linear relationship models, as shown in Fig. S11 in the *Supporting Materials*. The result shows that the model based on the latter has a better correlation. The uptake of OVOCs deserves further study.

The assumption is clarified in the manuscript:

Lines 205: "Note that we assume that gas-phase diffusion limitations of OVOCs were negligible, given their low uptake coefficients."

*3. Page 3, line 70, please also report measurement accuracy.*

Response: The uncertainty associated with the sensitivity of $C_2H_3O_2Cl$ arises from multiple factors, including the concentration of the calibration gas (2.76%), the flow and temperature control system (2%), wall losses in the calibration tube (5%), and variations due to relative humidity (2%). Additionally, accounting for the uncertainty in wall losses within the inlet tube (5%) and peak fitting during data processing (1%), the overall uncertainty in the measurement of $C_2H_3O_2Cl$ was estimated to be 11.1% using the basic formula for error propagation.

$$\frac{\sigma_f}{f} = \sqrt{(\frac{\sigma_a}{a})^2 + (\frac{\sigma_b}{b})^2}$$

where $\sigma_f$ is the error in the result $f = a \times b$ or $f = \frac{a}{b}$, and $\sigma_a$, $\sigma_b$ are the errors in a and b, respectively. As your comment, measurement accuracy was added.

Lines 73 – 74: "The total uncertainty of the measured chloroacetic acid was estimated to be 11.1%."

*4. Page 5, first paragraph: it seems that the correlations could be mostly attributed to diurnal variation, is this true? Considering the diurnal variation of oxidants, VOCs and other emissions, it is not surprised to have chloroacetic acid, aerosol mass, photolysis rate, etc. all peak in daytime. Similarly, correlation with RH could also be due to the diurnal variation of humidity.*

Response: To reduce the effect of diurnal variation on correlation analysis, the correlations between $C_2H_3O_2Cl$, meteorological factors, and chemical constituents from 10:00 – 14:00 during the two observation periods were examined, as shown in revised Fig. S1. Results show that inorganic reactive chlorine levels, photolysis rates and aerosol surface area concentrations and relative humidity can be regarded as related to CAA levels with correlation coefficient greater than 0.5 or less than -0.5 during 10:00 - 14:00, although the coefficients are smaller than those during all day.

[Figure]

Figure S1. The correlation coefficients between important meteorological factors and CAA concentration (a) during all day in 2020, (b) during all day in 2021, (c) during 10:00 – 14:00 in 2020 and (d) during 10:00 – 14:00 in 2021. $Cl_x = 2 \times Cl_2 + HOCl + BrCl$, and $Br_x = 2 \times Br_2 + BrCl$. All data are 1-h averages.

*5. Page 5, line 150-152: the energy barrier (~8 kcal mol-1) between IM1 and IM2 is unclearly shown in the 3-D relaxed scan (Fig. 2). Texts on the axes need to be clearer.*

Response: We have redrawn Fig. 2 for clarity.

[Figure]

Figure 2. Relaxed scan of Cl• addition to propene ($C_3H_6$). IM1 and IM2 are intermediates for Cl• addition to alpha-C and beta-C of propene, and CP1, CP2 and CP3 are the complexes derived from scans of IM1 and IM2 in terms of bond lengths (*r*) of alpha-C-Cl and beta-C-Cl as variables, respectively. Scanned potential energy surfaces of

IM1 (in blue) and IM2 (in green) take the total energy of the reactants Cl$^{\bullet}$ + propene as zero for reference. Energy in brackets in kcal mol$^{-1}$.

*6. According to Figure 1, 4, 6, and 8, it seems that the diurnal variation of Cl-OVOCs/chloroacetic acid can only be represented by adding a reactive uptake, and cannot be resolved by adding another gas formation pathway related to oxidants levels. Is this correct? If so, this could be evidence to support the importance of multiphase process. Also, does adding reactive uptake change the diurnal pattern of precursors?*

Response: Comparing the simulated results of chloroacetic acid and Cl-OVOCs in scenarios I – IV (Figs. 4 and 8), adding another gas formation pathway can increase their simulated levels but not change their simulated diurnal patterns. In scenario V, adding the reactive uptake of chloroacetic acid matches the simulated and observed pattern (Fig. 8), which can be seen as evidence of the atmospheric multiphase process. As shown in Figs. 4 and 6, adding reactive uptake also changes the diurnal pattern of Cl-OVOCs. We make Tab. 1 and Fig. 8 clearer for diurnal patterns, and the simulated and observed diurnal patterns of Cl-OVOCs and chloroacetic acid are discussed:

Lines 223 – 226: "As discussed in Section 3.2, the gas-phase chlorine chemistry of alkenes accounts for only 8% of the observed chloroacetic acid level and fails to explain the observed diurnal cycle. Adding reactive uptake of chloroacetic acid on aerosols aligns the simulated and observed daily variation, and including the heterogeneous source of chloroacetic acid increases the simulated level to 24 % – 48 % of the observed level (Fig. 8)."

**Table 1. Model scenarios**

| Scenario | Precursor | | | Uptake of Cl-OVOCs and chloroacetic acid[a] | Heterogeneous formation of chloroacetic acid[b] |
|---|---|---|---|---|---|
| | C$_2$H$_4$ | C$_3$H$_6$ | C$_5$H$_8$ | | |
| I | √ | | | | |
| II | | √ | | | |
| III | | | √ | | |
| IV | √ | √ | √ | | |
| V | √ | √ | √ | √ | |
| VI | √ | √ | √ | √ | √ |

[a]The reactive uptake of Cl-OVOCs on the aerosols is based on estimated uptake coefficients derived from QC calculations, excluding the heterogeneous formation of chloroacetic acid, and the γ value for chloroacetic acid is assumed to be the same as that for acetic acid (Wang et al., 2020). [b]The yield of chloroacetic acid from the reactive uptake of chloro-acetaldehyde is estimated as twice that of oxalic acid from aqueous-phase photochemical reactions involving glyoxal (Carlton et al., 2007).

[Figure]

Figure 8. Comparison of measured and simulated diurnal profiles of chloroacetic acid.

*7. Figure 8 has not been mentioned in the main text, please add it to the corresponding description.*

Response: It has been added to the last paragraph of section **3.3**.

*8. Figure 8: Why does adding reactive uptake (scenario VI) decrease chloroacetic acid level at night (compare to scenario IV).*

Response: This may result from uncertainties in the reactive uptake coefficients of Cl-OVOCs and other missing sources of chloroacetic acid. Budget analyses (Fig. R1) on scenario VI revealed that heterogeneous loss of chloroacetic acid reduces its simulated concentration at night. Previous research has reported that the reactive uptake coefficients of methylglyoxal increase as RH increases, while those of glyoxal and organic acids decrease at high RH (Chen et al., 2021; De Haan et al., 2018; Gen et al., 2018; Tong et al., 2010; Zeineddine et al., 2023). Adding RH-dependent reactive uptake coefficients to the model may improve the simulated diurnal pattern of chloroacetic acid at night. Chlorination of natural organic matter and other organic matter in the aqueous phase may also produce chloroacetic acid, which may act as the nighttime source of chloroacetic acid. The limitation discussion on our simulation results is added in the revised manuscript:

Lines 232 – 242: "With the aforementioned updates, the box model significantly improved its ability to simulate chloroacetic acid. However, there is still a discrepancy between the updated simulations and field measurements, which may result from the uncertainty in the parameters we use and other factors affecting chloroacetic acid. For example, previous studies have reported the RH dependence of the reactive uptake coefficients of aldehydes and organic acids (Chen et al., 2021; De Haan et al., 2018; Gen et al., 2018; Tong et al., 2010; Zeineddine et al., 2023). Other reactions of aldehydes in aerosols such as ˙OH oxidation, sulfite addition and Maillard-like reactions with reduced nitrogen species could compete with hydrolysis (Shen et al., 2024; Tang et al., 2022), potentially suppressing the yield of organic acids from the multiphase conversion of aldehydes. α-Chloro-β-ketones such as chloroacetone may also contribute to chloroacetic acid formation through atmospheric heterogeneous chemistry, supported by our QC calculations. In addition to alkenes, other VOCs of high molecular weight, such as ethylbenzene, may serve as precursors of chloroacetic acid (Cui et al., 2021). Moreover, chloroacetic acid may be produced as a disinfection by-product from the chlorination of dissolved organic matter in the aqueous phase (Jahn et al., 2024)."

[Figure]

Figure R1. Diurnal variation of chloroacetic acid production and loss rates on scenario VI in Table 1.

*9. I might suggest adding description for simulated contribution from heterogeneous reactions of chloro-acetaldehyde to the observed chloroacetic acid in abstract and conclusion.*

Response: It has been added.

Abstract:

"Box model simulation with multiphase chemistry reveals that the heterogeneous conversion of chloro-acetaldehyde to $C_2H_3O_2Cl$ can contribute 24% – 48% of the observed levels."

Conclusion:

"Adding the heterogeneous processes of these Cl-OVOCs to the MCM model explains 24% – 48% of the observed chloroacetic acid levels and exhibits a diurnal pattern similar to the observations, reducing the gaps between observed and simulated results."

*10. Abstract: "multiphase processes in VOC-Cl chemistry" in the last sentence may lead to confusion. It sounds like a multiphase process involve gas VOC with oxidation by Cl in aqueous phase. Please consider rephase.*

Response: We revise "multiphase processes in VOC-Cl chemistry" to "multiphase processes in atmospheric organic acid formation".

**Reference**

Carlton, A. G., Turpin, B. J., Altieri, K. E., Seitzinger, S., Reff, A., Lim, H.-J., and Ervens, B.: Atmospheric oxalic acid and SOA production from glyoxal: Results of aqueous photooxidation experiments, Atmos. Environ., 41, 7588–7602, https://doi.org/10.1016/j.atmosenv.2007.05.035, 2007.

Chen, X., Zhang, Y., Zhao, J., Liu, Y., Shen, C., Wu, L., Wang, X., Fan, Q., Zhou, S., and Hang, J.: Regional modeling of secondary organic aerosol formation over eastern China: The impact of uptake coefficients of dicarbonyls and semivolatile process of primary organic aerosol, Science of The Total Environment, 793, 148176, https://doi.org/10.1016/j.scitotenv.2021.148176, 2021.

Cui, H., Chen, B., Jiang, Y., Tao, Y., Zhu, X., and Cai, Z.: Toxicity of 17 Disinfection By-products to Different Trophic Levels of Aquatic Organisms: Ecological Risks and Mechanisms, Environ. Sci. Technol., 55, 10534–10541, https://doi.org/10.1021/acs.est.0c08796, 2021.

Davidovits, P., Kolb, C. E., Williams, L. R., Jayne, J. T., and Worsnop, D. R.: Update 1 of: Mass Accommodation and Chemical Reactions at Gas−Liquid Interfaces, Chem. Rev., 111, cr100360b, https://doi.org/10.1021/cr100360b, 2011.

De Haan, D. O., Jimenez, N. G., De Loera, A., Cazaunau, M., Gratien, A., Pangui, E., and Doussin, J.-F.: Methylglyoxal uptake coefficients on aqueous aerosol surfaces, J. Phys. Chem. A, 122, 4854–4860, https://doi.org/10.1021/acs.jpca.8b00533, 2018.

Gen, M., Huang, D. D., and Chan, C. K.: Reactive Uptake of Glyoxal by Ammonium-Containing Salt Particles as a Function of Relative Humidity, Environ. Sci. Technol., 52, 6903–6911, https://doi.org/10.1021/acs.est.8b00606, 2018.

Herrmann, H., Hoffmann, D., Schaefer, T., Bräuer, P., and Tilgner, A.: Tropospheric Aqueous-Phase Free-Radical Chemistry: Radical Sources, Spectra, Reaction Kinetics and Prediction Tools, ChemPhysChem, 11, 3796–3822, https://doi.org/10.1002/cphc.201000533, 2010.

Huang, Y., Xu, H., Chen, B., Pan, H., and Qiu, Z.: Insights into chloroacetaldehydes degradation by 254 nm ultraviolet: Kinetics, products, and influencing factors, Journal of Environmental Chemical Engineering, 9, 104571, https://doi.org/10.1016/j.jece.2020.104571, 2021.

Jahn, L. G., McPherson, K. N., and Hildebrandt Ruiz, L.: Effects of relative humidity and photoaging on the formation, composition, and aging of ethylbenzene SOA: Insights from chamber experiments on chlorine radical-initiated oxidation of ethylbenzene, ACS Earth Space Chem., 8, 675–688, https://doi.org/10.1021/acsearthspacechem.3c00279, 2024.

Jayne, J. T., Duan, S. X., Davidovits, P., Worsnop, D. R., Zahniser, M. S., and Kolb, C. E.: Uptake of gas-phase aldehydes by water surfaces, J. Phys. Chem., 96, 5452–5460, https://doi.org/10.1021/j100192a049, 1992.

Schweitzer, F., Magi, L., Mirabel, P., and George, C.: Uptake Rate Measurements of Methanesulfonic Acid and Glyoxal by Aqueous Droplets, J. Phys. Chem. A, 102, 593–600, https://doi.org/10.1021/jp972451k, 1998.

Shen, H., Huang, L., Qian, X., Qin, X., and Chen, Z.: Positive Feedback between Partitioning of Carbonyl Compounds and Particulate Sulfur Formation during Haze Episodes, Environ. Sci. Technol., 58, 21286–21294, https://doi.org/10.1021/acs.est.4c07278, 2024.

Tang, S., Li, F., Lv, J., Liu, L., Wu, G., Wang, Y., Yu, W., Wang, Y., and Jiang, G.: Unexpected molecular diversity of brown carbon formed by Maillard-like reactions in aqueous aerosols, Chem. Sci., 13, 8401–8411, https://doi.org/10.1039/D2SC02857C, 2022.

Tong, S. R., Wu, L. Y., Ge, M. F., Wang, W. G., and Pu, Z. F.: Heterogeneous chemistry of monocarboxylic acids on $\alpha$-Al$_2$O$_3$ at different relative humidities, Atmos. Chem. Phys., 10, 7561–7574, https://doi.org/10.5194/acp-10-7561-2010, 2010.

Wang, Y., Zhou, L., Wang, W., and Ge, M.: Heterogeneous Uptake of Formic Acid and Acetic Acid on Mineral Dust and Coal Fly Ash, ACS Earth Space Chem., 4, 202–210, https://doi.org/10.1021/acsearthspacechem.9b00263, 2020.

Zeineddine, M. N., Urupina, D., Romanias, M. N., Riffault, V., and Thevenet, F.: Uptake and reactivity of acetic acid on Gobi dust and mineral surrogates: A source of oxygenated volatile organic compounds in the atmosphere?, Atmos. Environ., 294, 119509, https://doi.org/10.1016/j.atmosenv.2022.119509, 2023.